

# Strategies for improved rhamnolipid production by *Pseudomonas aeruginosa* PA1

Alexandre Soares dos Santos[1], Nei Pereira Jr[2] and Denise M.G. Freire[3]

[1] Department of Basic Science/Faculty of Biological Science and Health, Universidade Federal dos Vales do Jequitinhonha e Mucuri, Diamantina, Minas Gerais, Brazil

[2] Department of Biochemical Engineering/School of Chemistry, Universidade Federal do Rio de Janeiro, Rio de Janeiro, Brazil

[3] Department of Biochemistry/Institute of Chemistry, Universidade Federal do Rio de Janeiro, Rio de Janeiro, Brazil

## ABSTRACT

Rhamnolipids are biosurfactants with potential for diversified industrial and environmental uses. The present study evaluated three strategies for increasing the production of rhamnolipid-type biosurfactants produced by *Pseudomonas aeruginosa* strain PA1. The influence of pH, the addition of *P. aeruginosa* spent culture medium and the use of a fed-batch process were examined. The culture medium adjusted to pH 7.0 was the most productive. Furthermore, the pH of the culture medium had a measurable effect on the ratio of synthesized mono- and dirhamnolipids. At pH values below 7.3, the proportion of monorhamnolipids decreased from 45 to 24%. The recycling of 20% of the spent culture medium in where *P. aeruginosa* was grown up to the later stationary phase was responsible for a 100% increase in rhamnolipid volumetric productivity in the new culture medium. Finally, the use of fed-batch operation under conditions of limited nitrogen resulted in a 3.8-fold increase in the amount of rhamnolipids produced ($2.9 \mathrm{~g~L}^{-1}$–$10.9 \mathrm{~g~L}^{-1}$). These results offer promising pathways for the optimization of processes for the production of rhamnolipids.

## INTRODUCTION

Rhamnolipids are biodegradable biological surfactants. They present low toxicity and high resistance to extreme conditions of pH, salinity and temperature (*Kesting et al., 1996*). Their surface properties, such as emulsification, dispersion, foaming, detergency, wetting and stabilization (*Van Dyke et al., 1993*; *Ishigami et al., 1994*; *Desai & Banat, 1997*; *Bognolo, 1999*), as well as their capacity for removing heavy metals (*Torrens, Herman & Miller-Maier, 1998*; *Lang & Wullbrandt, 1999*; *Kitamoto, Isoda & Nakahara, 2002*) and their anticorrosive capacities (*Araujo, Freire & Nitschke, 2013*), confer to these biosurfactants a variety of ecological (*Rahman et al., 2002*) and commercial applications in the oil, pharmaceutical, food and chemical industries (*Irfan-Maqsood & Seddiq-Shams, 2014*; *Randhawa & Rahman, 2014*; *Sinumvayo & Ishimwe, 2015*).

Rhamnolipids are mainly produced by *Pseudomonas aeruginosa*, a Gram-negative bacterium that can be isolated from various habitats (water, soil or even plants). The control

Corresponding author
Alexandre Soares dos Santos,
alexandreletam@gmail.com,
alexandre.soares@ufvjm.edu.br

of rhamnolipid production in *P. aeruginosa* is achieved by a regulatory system called quorum sensing that is controlled by autoinducers such as acyl homoserine lactones and Pseudomonas quinolone signal molecule. These signaling molecules, paired with the LasR and RhlR regulators, initiate the expression of the enzymes involved in rhamnolipid synthesis (rhamnosyltransferases) (*Ochsner et al., 1994*; *Ochsner, Hembach & Fiechter, 1995*; *Rahim et al., 2001*; *Reis et al., 2011*).

Efforts should be oriented toward the optimization of processes for the production of biosurfactants that result in high productivity on a commercial scale so that they can compete with synthetic surfactants in terms of cost. One of the strategies that has been suggested in the literature is the limitation of nutrients (*Desai & Banat, 1997*; *Chayabutra, Wu & Ju, 2001*), mainly nitrogen sources, as well as multivalent cations (*Syldatk et al., 1985*; *Glick et al., 2010*) and some anions (*Mulligan, Mahmourides & Gibbs, 1989*; *Clarke, Ballot & Reid, 2010*), as a condition necessary for stimulating the synthesis of rhamnolipids by *P. aeruginosa*. In addition to nutritional modifications, changes in physical factors such as temperature and pH can also influence the synthesis of rhamnolipids by *P. aeruginosa* (*Sousa et al., 2011*; *Jamal et al., 2014*). Another approach to the induction of the rhamnolipid synthesis is the use of exogenous or endogenous autoinducers (*Ochsner & Reiser, 1995*; *Nakata, Yoshimoto & Yamada, 1998*; *Galkin et al., 2014*).

The aim of this work was to improve the cultivation conditions for the production of rhamnolipids by a strain of *Pseudomonas aeruginosa* isolated from a Brazilian petroleum-exploring environment. This study involved the investigation of the effect of a variation in the pH of the culture medium, the medium supplementation with recycled *P. aeruginosa* spent culture medium and evaluation of process carried on fed-batch operation.

## MATERIAL AND METHODS

### Inoculum

*Pseudomonas aeruginosa* PA1 (*Santa Anna et al., 2001*) was maintained in a glycerol solution (10% v/v) at −80 °C. The thawed strain sample was inoculated onto YPDA plates (yeast extract, 0.3%; peptone, 1.5%; dextrose, 0.1%; agar, 1.2%) at 30 °C for 48 h. The growth of the inoculum was initiated by the addition of a loopful of cells from YPDA plates to a 1,000 mL Erlenmeyer flask containing 300 mL of medium with the following composition (per liter): 1.0 g of $NaNO_3$, 3.0 g of $KH_2PO_4$, 7.0 g of $K_2HPO_4$, 0.2 g of $MgSO_4.7H_2O$, 5.0 g of yeast extract, 5.0 g of peptone, and 30 g of glycerol. *P. aeruginosa* was grown at 30 °C and pH 7.0 in a rotary shaker at 170 rpm for 24 h. Cells were harvested by centrifugation (10,000 g for 30 min) and used as the inoculum.

### Influence of pH on rhamnolipid production

The culture medium contained (per liter) 0.2 g of $MgSO_47H_2O$, 1.38 g of $NaNO_3$, and 30 g of glycerol. The pH of the culture medium was adjusted from 5.7 to 8.0 with $KH_2PO_4/K_2HPO_4$ buffer by varying the mole fraction of salt species without changing the concentration of total phosphate ion, which was maintained at 0.062 moles per liter. Six 500-mL Erlenmeyer flasks containing a working volume of 100 mL were prepared, inoculated with 0.12 g of cells (dry weight) and incubated at 30 °C in a rotary shaker at

170 rpm for 192 h. Samples were removed at 24-hour intervals for the measurement of cell growth and rhamnolipid concentration.

## Simple batch process supplemented with recycled *P. aeruginosa* spent culture medium

The culture medium was prepared with the following composition (per liter): 1.38 g of $NaNO_3$, 3.0 g of $KH_2PO_4$, 7.0 g of $K_2HPO_4$, 0.2 g of $MgSO_4.7H_2O$, and 30 g of glycerol. To each of five 500-mL Erlenmeyer flasks was added 100 mL of liquid medium supplemented with 1%, 5%, 10%, 15% or 20% (v/v) of cell-free spent culture supernatant from a 120 h-old *P. aeruginosa* PA1 culture. The control flask contained no spent culture medium. The flasks were inoculated with 0.05 g of cells (dry weight) and incubated at 30 °C and pH 7.0 in a rotary shaker at 170 rpm for 200 h. Aliquots (1.5 mL) were removed at 24-hour intervals for the assessment of cell growth and rhamnolipid production. The mass balance (carbon and nitrogen sources) present at the start of the culture process was adjusted, when necessary, to maintain the same C/N ratio and avoid undesirable side effect.

## Fed-batch process: nitrogen and carbon feed

The culture medium contained (per liter) 3.0 g of $KH_2PO_4$, 7.0 g of $K_2HPO_4$, 0.2 g of $MgSO_4.7H_2O$, 0.46 g of $NaNO_3$, and 10.0 g of glycerol. The pH was adjusted to 7.0. Three 500-mL Erlenmeyer flasks (A, B and C) were prepared containing 100 mL of the liquid medium. The flasks were inoculated with 0.2 g of cells (dry weight) and incubated at 30 °C in a rotary shaker at 170 rpm for 240 h. The frequency of addition of nutrients was a function of the amount of glycerol consumed in each flask. Flask A received 5 mL of a solution containing glycerol ($200\ g\ L^{-1}$) and sodium nitrate ($9\ g\ L^{-1}$) at each addition. Flask B received 5 mL of a solution containing only glycerol ($200\ g\ L^{-1}$), and flask C received 5 mL of sterile distilled water. Aliquots (1.5 mL) were removed at 24-hour intervals for the quantification of cell growth, nitrate and glycerol consumption and rhamnolipid production.

## Thin-layer chromatography for analysis of rhamnolipid types

The rhamnolipids produced were extracted with ethyl acetate from acidified (pH 2.0) cell-free medium and analyzed by TLC on silica-gel-coated aluminum sheets (Macherey-Nagel®) using $CHCl_3:CH_3OH:CH_3COOH$ (65:15:2) as the eluent (*Schenk, Schuphan & Schmidt, 1995*). The separated zones were stained with orcinol-sulfuric acid reagent, followed by heating at 100 °C for 15 min. The stained plates were scanned, and the relative quantities of the spots corresponding to the monorhamnolipids and dirhamnolipids were determined by densitometry using Band Leader (Ma'ayan Aharoni) and Micronal Origin (Micronal Software, Inc.) software to produce two dimensional chromatograms.

## Determination of cell, rhamnolipid, glycerol and nitrate concentrations

Cell growth was assessed by measuring the absorbance at 500 nm, and the cell dry weight ($g\ L^{-1}$) was determined using a standard calibration curve [$ABS = 1.2595 \times DW\ (g\ L^{-1}) - R^2 = 0.989$], considered to be valid for absorbance values up to 0.6 OD. Rhamnolipid quantification was achieved indirectly by measurement of the rhamnose concentration

using the method of *Dubois et al. (1956)* and was expressed as the rhamnolipid concentration using the factor 2.23 established by *Kronemberger et al. (2007)* by mass spectrometry measurements. A 0.5 mL volume of cell-free supernatant was mixed with 0.5 mL of 5% phenol solution and 2.5 mL of 98% sulfuric acid and incubated for 15 min before measuring the absorbance at 490 nm. The results were compared with the analytical curve for rhamnose. Glycerol was quantified by the GPO-POD enzymatic-colorimetric method using a kit for triglyceride determination from LaborLab® (Brazil). Nitrate was quantified through a colorimetric method using brucine sulfate (*ACS, 2006*). Briefly, 2 mL of 0.6 g L$^{-1}$ brucine sulfate in sulfuric acid solution (80%) was added to 0.5 mL of sample, and the reaction mixture was heated in boiling water for 15 min. The reaction mixture was immediately cooled in an ice bath, and the absorbance was measured at 410 nm. The absorbance values were converted into concentration using an analytical curve for sodium nitrate.

## Definition of process parameters utilized

The process parameters utilized to evaluate the progress of improvement strategies for rhamnolipid production are defined as follows. $t_f$: final time of process (h); $t_i$: initial time of process (h); $\Delta t$ $(t_f - t_i)$; $P_f$: final rhamnolipid concentration (g L$^{-1}$); $P_i$: initial rhamnolipid concentration (g L$^{-1}$); $X_i$: initial cell mass concentration (g L$^{-1}$); $X_f$: final cell mass concentration (g L$^{-1}$); $S_f$: final substrate concentration (g L$^{-1}$); $S_i$: initial substrate concentration (g L$^{-1}$); $\Delta$Rhamnolipids $(P_f - P_i)$; $\Delta$Biomass $(X_f - X_i)$; $\Delta$Substrate $(S_i - S_f)$; $Y_{P/X}$ ($\Delta$Rhamnolipids $\div$ $\Delta$Biomass): yield of product synthesized per unit of cell mass produced (g g$^{-1}$); $Y_{P/S}$ ($\Delta$Rhamnolipids $\div$ $\Delta$Substrate): yield of product synthesized per unit of substrate consumed (g g$^{-1}$); $Q_P$ ($\Delta$Rhamnolipids $\div$ $\Delta t$): volumetric rhamnolipid production rate (g L$^{-1}$ h$^{-1}$); $Q_{S(Gly)}$ ($\Delta$Substrate $\div$ $\Delta t$): volumetric glycerol consumption rate (g L$^{-1}$ h$^{-1}$); $Q_{S(NO3)}$ ($\Delta$Substrate $\div$ $\Delta t$): volumetric nitrate consumption rate (g L$^{-1}$ h$^{-1}$); $q_P$ ($Y_{P/X} \times 1,000 \div \Delta t$): specific rate of rhamnolipids synthesis (mg g$^{-1}$ h$^{-1}$).

# RESULTS AND DISCUSSION

## Influence of pH on rhamnolipid synthesis

The qualitative effect of pH on rhamnolipid synthesis was evaluated by thin layer chromatography. Densitometric analysis of mono- and dirhamnolipids performed by TLC (Fig. 1) furnished relative migration values close to 0.8 for the monorhamnolipids and 0.5 for the dirhamnolipids. These values are in agreement with the migration rate of monorhamnolipids and dirhamnolipids observed by Schenk and collaborators (*1995*), who employed TLC analysis under the same conditions. Under the assay conditions, the more hydrophilic dirhamnolipids interact more strongly with the TLC (silica gel) stationary phase because of the presence of two rhamnose rings linked to lipid chain, whereas only one sugar ring exists in the monorhamnolipids species.

As is shown in Table 1, the percentage of mono- and dirhamnolipids, determined by densitometry of TLC plates, varied as a function of the pH of the culture medium. For pH $\leq 7.0$, the amount of monorhamnolipids produced was lower than the amount of dirhamnolipids. On the other hand, at pH values higher than 7.0, the amounts of mono- and

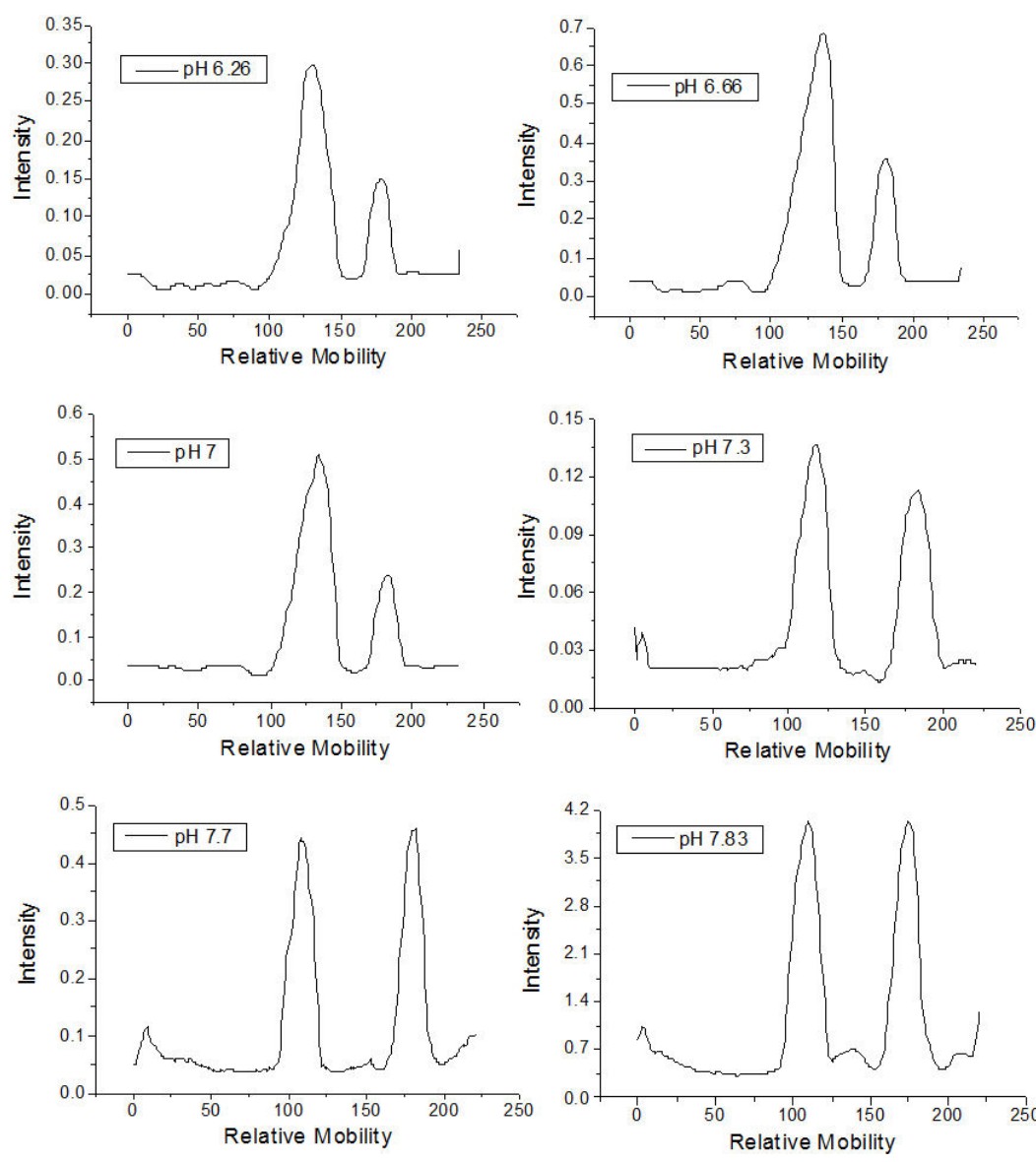

**Figure 1** **Densitometry of the thin layer chromatographs of rhamnolipids synthesized in culture medium at different pH values.**

dirhamnolipids were approximately the same. *Mata-Sandoval, Karns & Torrents (2001)* showed that, at pH 6.26 and 7.0, dirhamnolipids were the most abundant molecular species in the mixtures of rhamnolipids synthesized by *P. aeruginosa* with corn oil as sole carbon source. However, they limited their experiments up to pH 7.0.

It is possible that pH acts at a transcriptional level on the synthesis of one of the rhamnosyltransferases or on the synthesis of a specific glycolipid transporter channel, or even on the synthesis of L-rhamnose in *P. aeruginosa* (*Olvera et al., 1999*). *Escherichia coli* represents an example in which various enzymes and periplasmic proteins are expressed in a pH-dependent mechanism (*Stancik et al., 2002*). In any case, the possibility of obtaining

**Table 1** Relative percentages of the rhamnolipid types synthesized as a function of the pH of the culture medium.

| pH | Dirhamnolipids (%) | Monorhamnolipids (%) |
| --- | --- | --- |
| 6.26 | 74.35 | 25.65 |
| 6.66 | 75.35 | 24.65 |
| 7.00 | 76.41 | 23.59 |
| 7.30 | 55.25 | 44.75 |
| 7.70 | 51.66 | 48.34 |
| 7.83 | 52.75 | 47.25 |

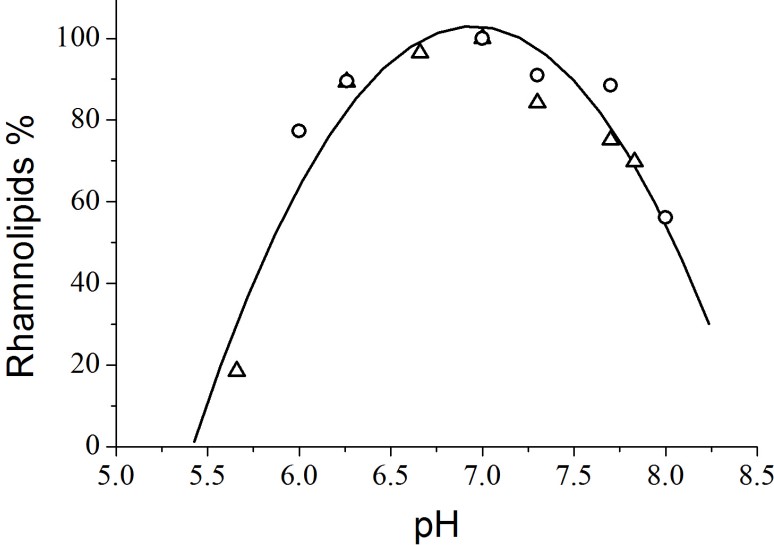

**Figure 2** Variation of rhamnolipid synthesis as a function of the pH of the culture medium. The triangles and circles correspond to independent experiments.

a product with distinct percentages of rhamnolipid types synthesized by the appropriate choice of pH allows for different uses and applications for this natural formulation. Indeed, different rhamnolipid types or their mixtures present emulsifying activity, critical micellar concentration, hydrophobicity or bioavailabilities distinct from one other (*Mata-Sandoval, Karns & Torrents, 1999*). *Costa et al. (2010)*, for example, observed different CMC, emulsifying activity and interfacial tension values when they compared two rhamnolipid preparations with distinct proportions of mono and dirhamnolipids species.

From the viewpoint of quantitative analysis, the effect of the pH of the culture medium on total rhamnolipid synthesis by *P. aeruginosa* PA1 demonstrated that the most productive pH value was 7.0 (Fig. 2). *Guerra-Santos, Käppeli & Fiechter (1986)* found that rhamnolipid production by *P. aeruginosa* cultivated in glucose peaked at pH values ranging from 6.0 to 6.5 and decreased at values higher than 7.0. *Jamal et al. (2014)* achieved a maximum rhamnolipids yield (4.44 g L$^{-1}$) at a pH level of 7.33 using glycerol as the carbon source and NaNO$_3$ as the nitrogen source.

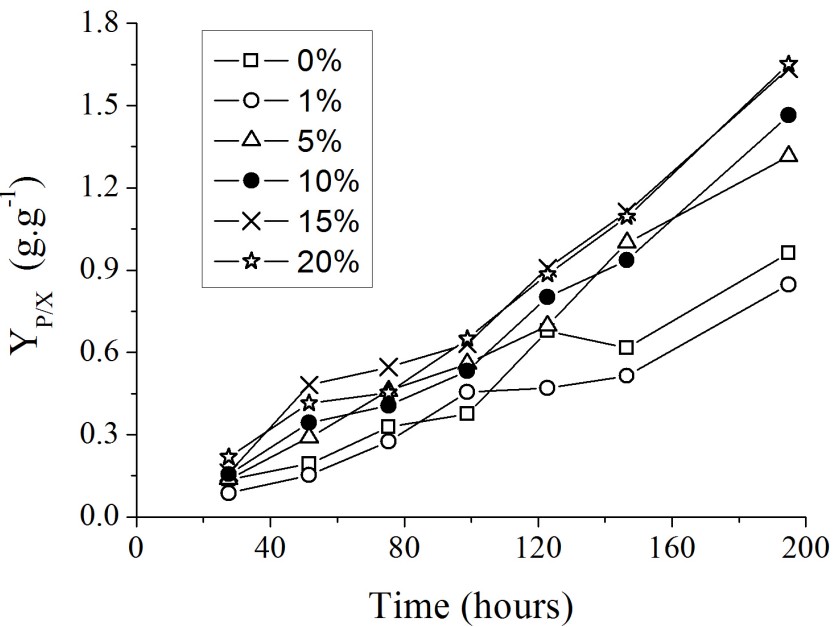

**Figure 3** Progress curves of yield coefficient $Y_{P/X}$ in culture medium with different supplemental proportions of 120 h-old *P. aeruginosa* spent culture medium.

## Use of recycled *P. aeruginosa* spent culture medium

In this approach, the free cell spent cultivation medium, where *P. aeruginosa* was grown for a 120-hour period (later stationary phase), was used to induce rhamnolipid production. The presence of a mixture of acylated homoserine lactones (AHLs) and Pseudomonas quinolone signal (PQS) naturally secreted by *P. aeruginosa* in later stationary growth phase was reported by several researchers (*Ochsner & Reiser, 1995*; *Pesci et al., 1997*; *Reis et al., 2011*). This autoinducers are known to be responsible for the induction of the synthesis of rhamnolipids and other virulence factors in *P. aeruginosa* (*Fuqua & Greenberg, 1998*; *Williams et al., 2000*; *Diggle et al., 2003*).

The addition of spent medium containing endogenous autoinducers at the start of new *P. aeruginosa* cultures resulted in an increase in the yield of rhamanolipids synthesized per unit of substrate consumed ($Y_{P/X}$), as showed in Fig. 3. The most significant differences among the $Y_{P/X}$ values were observed when the spent culture medium concentrations were changed from 1% to 5% and from 5 to 15% (Table 2).

The addition of spent culture medium to the new *P. aeruginosa* PA1 culture also increased the volumetric glycerol consumption rate ($Q_{S(Gly)}$) (Table 2). This fact suggests that the addition of endogenous autoinducers would also alter the metabolic rate of nutrients that, in the final analysis, would be coupled to the synthesis of the precursors for rhamnolipid production. The volumetric rhamnolipids production rate ($Q_P$) was also favored by the addition of endogenous autoinducers (Table 2).

The slopes calculated for each curve shown in Fig. 3 using liner regression, called specific rate of rhamnolipids synthesis, were plotted against the log values of the spent culture medium concentration and are presented in Fig. 4. This graph permits one to clearly

**Table 2** Process parameters of rhamnolipid production by *P. aeruginosa* PA1 in culture medium supplemented with different percentages of spent medium containing endogenous autoinducers.

| Parameters | 0% | 1% | 5% | 10% | 15% | 20% |
|---|---|---|---|---|---|---|
| $\Delta$Rhamnolipids (g L$^{-1}$) | 4.79 | 4.50 | 7.85 | 8.09 | 8.83 | 9.39 |
| $\Delta$Biomass (g L$^{-1}$) | 4.98 | 5.31 | 5.96 | 5.53 | 5.40 | 5.68 |
| $Y_{P/X}$ (g g$^{-1}$) | 0.96 | 0.85 | 1.32 | 1.46 | 1.63 | 1.65 |
| $Y_{P/S}$ (g g$^{-1}$) | 0.43 | 0.33 | 0.32 | 0.32 | 0.36 | 0.38 |
| $Q_P$ (g L$^{-1}$ h$^{-1}$) | 0.027 | 0.026 | 0.044 | 0.047 | 0.051 | 0.054 |
| $Q_S$ (g L$^{-1}$ h$^{-1}$) | 0.014 | 0.012 | 0.021 | 0.028 | 0.029 | 0.031 |

**Notes.**

$\Delta$Rhamnolipids, Difference between final and initial rhamnolipid concentration; $\Delta$Biomass, Difference between final and initial cellular concentration; $Y_{P/X}$, Yield of product synthesized per unit of cell mass produced; $Y_{P/S}$, Yield of product synthesized per unit of substrate consumed; $Q_P$, Volumetric rhamnolipids production rate; $Q_{S(Gly)}$, Volumetric glycerol consumption rate.

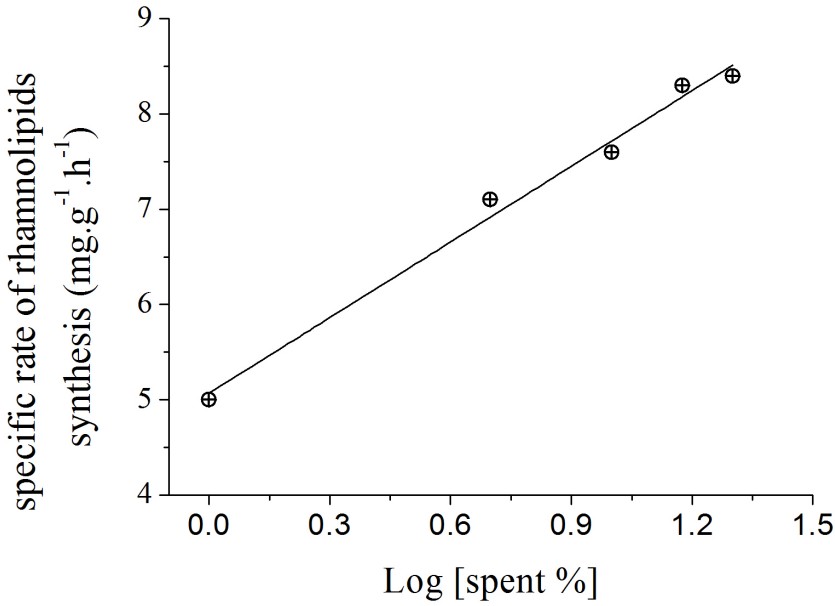

**Figure 4** Relation between the specific rate of rhamnolipid synthesis and the addition of different proportions of 120 h-old *P. aeruginosa* spent culture medium.

observe that the increase in the production of rhamnolipids was due to an increase in the capacity of the microorganisms to synthesize rhamnolipids. These results suggest that the amount of spent culture medium added at the beginning of cultivation probably contributed to the increase in the synthesis of rhamnosiltransferases because the natural unfolding of the quorum sensing system, which is responsible for the transcriptional regulation of rhamnolipid synthesis in presence of endogenous autoinducers.

## Fed-batch strategy

The fed-batch operation is another possible strategy for increasing the production of rhamnolipids and was chosen to circumvent a possible negative effect, observed by *Santa Anna et al. (2002)*, on the assimilation of nutrients when the culture medium contains glycerol

**Table 3  Process parameters observed in fed batch processes under different feeding conditions.**

| Parameters | C + N | C | $H_2O$ |
|---|---|---|---|
| $\Delta$Rhamnolipids (g L$^{-1}$) | 10.93 | 7.76 | 2.88 |
| $\Delta$Biomass (g L$^{-1}$) | 4.78 | 3.34 | 2.41 |
| $Y_{P/X}$ (g g$^{-1}$) | 2.29 | 2.32 | 1.19 |
| $Y_{P/S}$ (g g$^{-1}$) | 0.33 | 0.35 | 0.21 |
| $Q_P$ (g L$^{-1}$ h$^{-1}$) | 0.047 | 0.033 | 0.027 |
| $Q_{S(Gly)}$ (g L$^{-1}$ h$^{-1}$) | 0.168 | 0.073 | 0.197 |
| $Q_{S(NO3)}$ (g L$^{-1}$ h$^{-1}$) | 0.027 | 0.027 | 0.027 |

**Notes.**

$\Delta$Rhamnolipids, Difference between final and initial rhamnolipid concentration; $\Delta$Biomass, Difference between final and initial cellular concentration; $Y_{P/X}$, Yield of product synthesized per unit of cell mass produced; $Y_{P/S}$, Yield of product synthesized per unit of substrate consumed; $Q_P$, Volumetric rhamnolipids production rate; $Q_{S(gly)}$, Volumetric glycerol consumption rate; $Q_{S(NO3)}$, Volumetric nitrate consumption rate. The parameters for the condition "$H_2O$" were calculated at 100 h of culture.

concentrations higher than 3%. In addition, a process driven by fed-batch can control and maintain the nutrient limits already established as being favorable for rhamnolipid synthesis (*Desai & Banat, 1997*; *Chayabutra, Wu & Ju, 2001*; *Xavier, Kim & Foster, 2011*).

The consumption of nutrients during the fed-batch process was determined. In the control experiment, in which only water was fed to the medium, the carbon source (glycerol) and nitrogen source (NaNO$_3$) present at the beginning were totally consumed after approximately 50 and 24 h of cultivation, respectively (Figs. 5A and 5B).

Although the consumption of the nitrogen source (NaNO$_3$) occurred in shorter time intervals than glycerol consumption (Figs. 5A and 5B), the addition of a solution containing glycerol and nitrate was performed as a function of the consumption of the carbon source (Fig. 5A). At 143 h after the beginning of the process, the volumetric glycerol consumption rate decreased from 195 mg L$^{-1}$ h$^{-1}$ to 162 mg L$^{-1}$ h$^{-1}$. Because of this decay, only sodium nitrate was added at this time to the flask originally fed with carbon and nitrogen sources with the objective of maintaining the frequency of nitrate addition. The volumetric glycerol consumption rate ($Q_{S(Gly)}$) was higher in the presence of a nitrogen source (Table 3). This fact becomes apparent when the curves of glycerol consumption are compared with one another, considering the process in which glycerol and nitrate are added together and the process in which only glycerol was added (Fig. 5A). It is reasonable to assume that the enzymatic machinery involved in glycerol metabolism, as well as in rhamnolipid biosynthesis, depends on the assimilation of nitrogen and its conversion into catalytic proteins.

*Ochsner, Hembach & Fiechter (1995)* observed that the activity of the rhamnosyltransferase in *P. aeruginosa* during cultivation in a nitrogen-limiting medium containing glycerol as the carbon source was the highest at the beginning of the stationary phase and declined to zero in the late stationary phase. We suggest that exhaustion of the nitrogen source would limit not only cellular growth, but also the maintenance of the enzymatic machinery, mainly that involved in the metabolic pathways for rhamnolipid synthesis. In the present work, when the process of rhamnolipids production was simultaneously fed with carbon and nitrogen sources, a higher cell yield and volumetric productivity of rhamnolipids was achieved (Table 3).

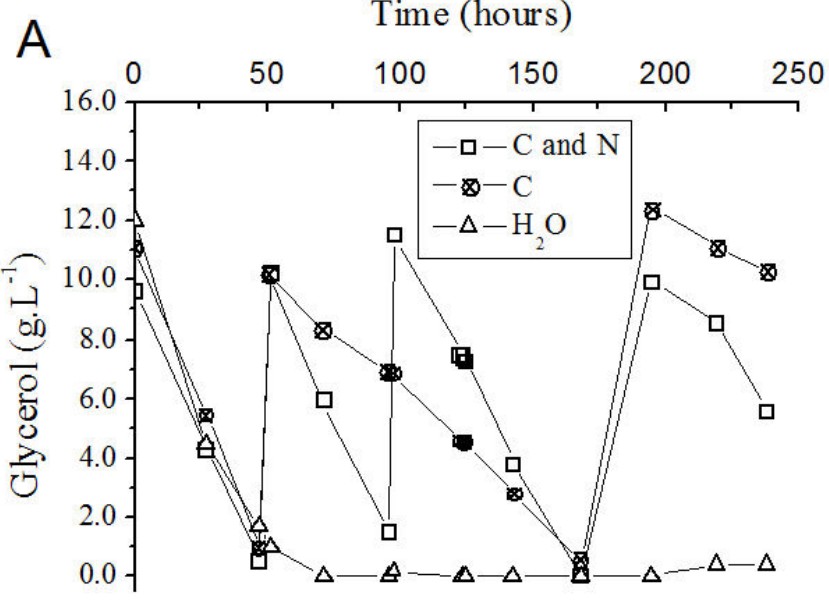

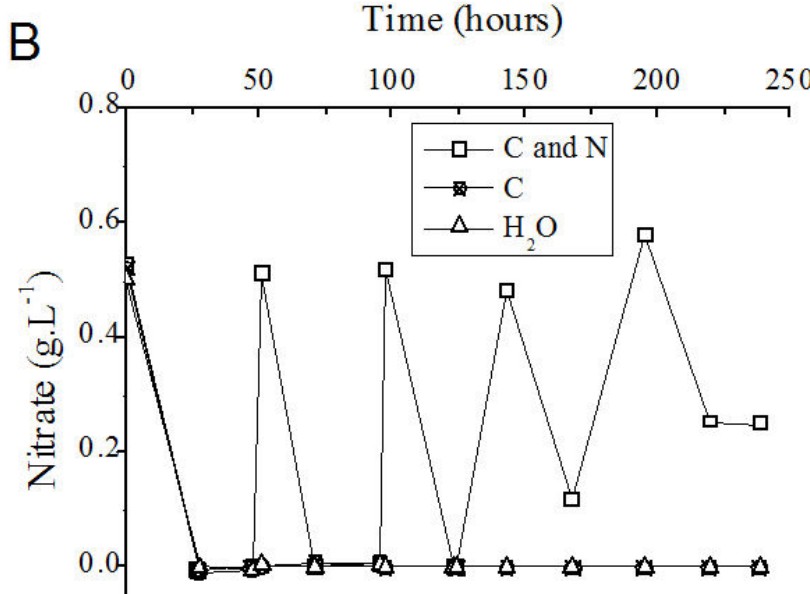

**Figure 5 Progress curve of glycerol (A) and nitrate (B) consumption during the fed-batch process using different feeding strategies.** C and N, fed with carbon and nitrogen sources together. C, fed only with the carbon source. H₂O, fed with water instead of nutrients.

On the basis of the feeding conditions, together with the control experiment (without feeding), one can conclude that the exhaustion of carbon and nitrogen sources interrupted the *P. aeruginosa* growth (Fig. 6A) and the rhamnolipid synthesis (Fig. 6B) sooner (at 72 h after initiating the cultivation). On the other hand, feeding the system with only glycerol led to an improvement when compared with the simple batch (Fig. 6A) and resulted in higher

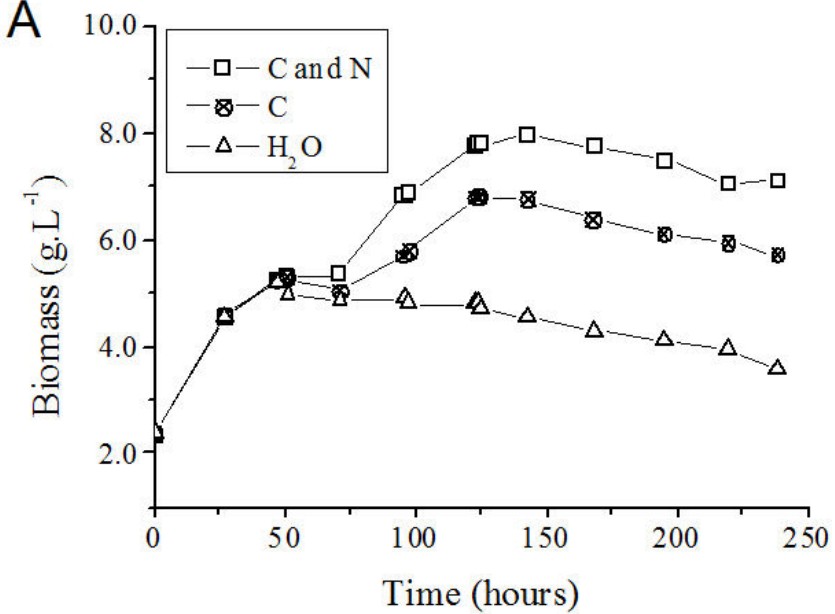

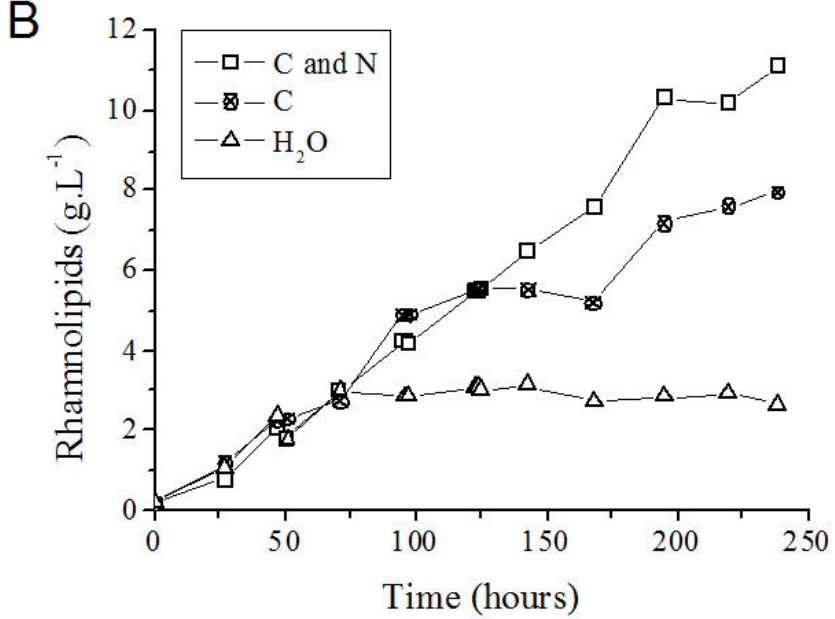

**Figure 6** *P. aeruginosa* **growth curves (A) and time course of rhamnolipid production (B) in the fed-batch process using different feeding strategies.** C and N, fed with carbon and nitrogen sources together. C, fed only with the carbon source. $H_2O$, fed with water instead of nutrients.

$Y_{P/X}$ and $Y_{P/S}$ values (Table 3). Furthermore, the feeding condition with both sources of nutrients (carbon and nitrogen) resulted in an even greater increase in both the biomass produced (Fig. 6A) and the volumetric productivity, reaching a rhamnolipid production value 40% higher (Table 3).

After 125 h from the start of the process, cell growth stopped in the fed-batch system with both sources of nitrogen and carbon and in the system only fed with carbon source

(Figs. 6A and 6B). It is possible that the limitation of other nutrients has occurred at this point, such as trace elements or oxygen. However, the production of rhamnolipids remained active mainly when fed simultaneously with nitrate and glycerol. This behavior characterizes a semi-growth associated profile.

## CONCLUSION

In addition to demonstrating the most appropriate pH for stimulating the production of rhamnolipids by *P. aeruginosa* PA1, selective synthesis of different types of rhamnolipids caused by certain pH ranges appeared as an unexpected and timely result. The reutilization or recycling of culture medium containing endogenous autoinducers of the quorum sensing system produced by *P. aeruginosa* in new culture medium for the production of rhamnolipids was very effective in the induction of rhamnolipid synthesis. A two-fold increase in volumetric productivity was obtained using this strategy. The fed-batch experiment using the limitation of the carbon and nitrogen source was successful and could be employed along with an appropriate formulation of the cultivation conditions (micronutrients, pH, supplementary autoinducers).

### Funding
This work was financially supported by the Agência Nacional do Petróleo, Gás Natural e Biocombustíveis (ANP), Financiadora de Estudos e Projetos (FINEP), Fundação Carlos Chagas Filho de Amparo à Pesquisa do Estado do Rio de Janeiro (FAPERJ) and Conselho Nacional de Desenvolvimento Científico e Tecnológico (CNPq). The funders had no role in study design, data collection and analysis, decision to publish, or preparation of the manuscript.

### Grant Disclosures
The following grant information was disclosed by the authors:
Agência Nacional do Petróleo, Gás Natural e Biocombustíveis (ANP).
Financiadora de Estudos e Projetos (FINEP).
Fundação Carlos Chagas Filho de Amparo à Pesquisa do Estado do Rio de Janeiro (FAPERJ).
Conselho Nacional de Desenvolvimento Científico e Tecnológico (CNPq).

### Competing Interests
The authors declare there are no competing interests.

### Author Contributions
- Alexandre Soares dos Santos conceived and designed the experiments, performed the experiments, analyzed the data, wrote the paper, prepared figures and/or tables, reviewed drafts of the paper.
- Nei Pereira Jr analyzed the data, contributed reagents/materials/analysis tools.
- Denise M.G. Freire conceived and designed the experiments, analyzed the data, contributed reagents/materials/analysis tools.

## Data Availability

The raw data has been supplied as Data S1.

## Supplemental Information

Supplemental information for this article can be found online at http://dx.doi.org/10.7717/peerj.2078#supplemental-information.

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
