# Peer review of "Strategies for improved rhamnolipid production by Pseudomonas aeruginosa PA1"

_PeerJ, doi:10.7717/peerj.2078_

## Round 0.1 · original submission · Major Revisions

· Academic Editor

Major Revisions

Dear Authors,

You are requested to go through the comments made by the reviewers (in particular those of Reviewer 2) and undertake the possible revisions to your Mss.

Looking forward to see your revised Mss.

Best regards,
Vijai K Gupta

Reviewer 1 ·

Basic reporting

The references used in this article are relatively too old.

Experimental design

No Comments。

Validity of the findings

No Comments

Additional comments

Comments to the Author
Manuscript: Strategies for improved rhamnolipid production by Pseudomonas aeruginosa PA1

In this study, authors investigated three strategies to increase the production of rhamnolipid produced by Pseudomonas aeruginosa strain PA1. This is an interesting report on improvement of rhamnolipid production. Different initial pH leads to different types of rhamnolipids. The reutilization of culture medium containing endogenous homoserine lactones significantly improved the rhamnolipid production. The fed batch strategy also enhanced rhamnolipid production by using conditions of carbon and nitrogen source limitation.
As presented, there are some issues with the report. The following are my detailed comments.

1. The references used in this article are relatively too old. Some recent published articles need cite.

2. For the strategy of adding endogenous autoinducers, in this study, 20% of the spent culture medium where P. aeruginosa was grown up to the later stationary phase was added. I wonder that how to eliminate the disturbance of the rhamnolipid product also synthesized in the present culture medium.

Reviewer 2 ·

Basic reporting

The manuscript deals with the optimization of rhamnolipids production using glycerol as carbon substrate. Three approaches are presented and productivities compared. At the end of the work, improved results are presented, however the partial conclusions are not integrated in the dessign of the procces production

Experimental design

Overall the present manuscript lacks of detailed description of the experiments (feed batch) and the interpretation of the results, and part of the information is repeated (pH effect)

Validity of the findings

The findings of the present work lay on the strategy of rhamnolipid production although authors claim they have three stratyegies to increase production yield, they did not applied them into the process, thus the work appears unfinished

Additional comments

Ref. manuscript 2015:09:6806:0:1:New 20 Oct 2015


Other comments
Figures: the legends of the figures should contain enough details of the culture conditions of the assay to easily understand the results presented.
Tables: Δ Biomass or Δ Rhamnose, does meant the increase or bacterial growth minus the RL content of the supernatant added? No information is supply in M&M section about the calculation of the rhamnose of the cultures.
Figure 4. Clarify the legend, what do you meant with specific rate of rhamnolipid production? As plotted in the graph mg/g.h; is that mg of product per g of cells per hour? Be careful the legend of the y axis says rhamnolipids it is so? Or it should read rhamnose? On the text is is said that the graph permits clearly observe that the production do not depends on biomass.
Fig, 5 and Fig. 6: in these figures a discontinuous addition is shown therefore no identical lines should be drawn from the lower point to the highest point.

Results and discussion
- Information of Figure 1, and Table 1 is similar on my opinion I would delete Fig. 1, Table 1 gives more detailed information.

- Authors adjusted the pH of the culture with phosphate buffer and hence a variation of the phosphate content in the culture medium. It is known that production is affected by that nutrient. What is the rational to conclude the production is only affected by the pH?

- Lines 159-161, tighgt some of the physic-chemical properties such as cmc or ST varies with the composition but in other cases ex. stability in front salts or emulsifying activity despite the differences in hydrophobicity are similar

- Lines 162-168. Regarding to the effect of pH on production, it is said on M&M section that the pH was adjusted by buffer phosphate from pH 6-8. Two questions: in Figure 2 it is shown pH 5.5 (?). It is known that RL production is negatively affected by phosphate concentration. No information about the content of phosphate is mentioned neither the growth of the population, since at pH 5.5 -6 the growth of Pseudomonas might be affected, moreover what is the cellular yield of RL production?

- Lines 217-179. It is said that NaNO3 concentration over 0.1% (1 g/L) inhibits the growth of Pseudomonas, I do not agree with this general appreciation, it might depend on the strain; there are numerous examples using higher amount of sodium nitrate and the bacterial growth increases with the nitrate content.

- Regarding to the addition of AHL, in the form of supernatant, the authors are right AHL has effect on RhlR production however, I wonder whether authors have considered that in the supernatant contain also rhamnolipids, do rhamnolipids have any effect?

- the profile of the bacterial growth should be provide together with nutrient addition in Fig. 5. As presented culture was feed when nutrients, (carbon or nitrate) were exhausted, what happened with the bacterial growth? When the stationary phase started? Since the assimilation of nitrate did decrease after 150 hours of culture (Fig. 6). Does the production yield change along the feed-batch culture?
What is the pH of the culture? and how do you control it? no information is provided.

---

## Round 0.2 · accepted · Accept

· Academic Editor

Accept

I have read your rebuttal and revised ms and believe it is up to the standard of the journal. I recommend the same for publication in PeerJ.